# LoLaMeMe: Logic, Language, Memory, Mechanistic Framework

## Abstract

The performance of Large Language Models have achieved superhuman breadth with unprecedented depth. At the same time, the language models are mostly black box models and the underlying mechanisms for performance have been evaluated using synthetic or mechanistic schemes. We extend current mechanistic schemes to incorporate Logic, memory, and nuances of Language such as latent structure. The proposed framework is called LoLaMeMe and we provide two instantiations of LoLaMeMe: LoLa and MeMe languages. We then consider two generative language model architectures: transformer-based GPT-2 and convolution-based Hyena. We propose the hybrid architecture T Hex and use LoLaMeMe framework is used to compare three architectures. T Hex outperforms GPT-2 and Hyena on select tasks.

## 1 Introduction

Language and language understanding have several nuances such as grammar, types, memory, reasoning, logic, short-term memory, and long-term memory. For example, consider the three sentences "Albert is a famous physicist", "Albert Fermi is a famous physicist", "Albert Fermi is a famous physicist in the new book Everywhere Nowhere". Each of the sentences constitutes the short-term memory (or prompt, in Language Model parlance). Understanding of these sentences requires interaction with the long-term memory or fact store. The first sentence is likely to match with "Albert Einstein" while the second one is likely to match "Albert Einstein" and "Enrico Fermi". The last sentence is most likely to be interpreted in its own context and it may not have matches. One can consider long-term memory[1] to be shaped by pre-training and fine-tuning while short-term memory is shaped by prompts and in-context learning. While pre-training. fine-tuning, in-context learning, and prompting have made it to the mainstream, the mechanistic understanding of these are hard given the mammoth sizes of datasets used for training the models and ever-lingering doubts of train-test overlap.

In addition, different Language Model architectures capture different aspects of language and frameworks to systematically compare them do not exist. The emerging framework of mechanistic evaluation is very promising. Our goal is to contribute to mechanistic evaluation research with a new framework which is capable of mimicking the many nuances of natural language. Our framework, called LoLaMeMe, supports the following aspects of natural language: word sizes, permanent vs temporary facts, latent types, and noise.

We then use the LoLaMeMe framework to explore the capabilities of different language model architectures. Transformer-based architectures are most widely used for language models and there are emerging convolution-based architectures such as Hyena. We evaluate the performance of these models using LoLaMeMe and propose a hybrid architecture, called T Hex, which performs better on certain aspects of natural language.

Our contributions are are as follows.

1. We propose a framework called LoLaMeMe which is a mechanistic framework closer to natural language.

---

[1]The uses of terms *short-term* and *long-term* in the introduction are figurative and not literal or technical. The main contributions do not depend on literal or metaphorical interpretations.

2. We create multiple datasets based on LoLaMeMe consisting of several billion tokens.

3. We evaluate two popular architectures GPT-2 and Hyena and show the architectures have complementary strengths.

4. We propose a new hybrid architecture, T Hex that is based on GPT-2 and Hyena. T Hex outperforms GPT-2 and Hyena on most LoLaMeMe tasks and on a related benchmark dataset.

## 2 Related Work

This research touches upon two lines of recent work: mechanistic interpretation and language model architectures.

### 2.1 Mechanistic Interpretation

Mechanistic Interpretability is the reverse-engineering of neural network models into human understandable concepts Olah (2022). There is lot of recent research interest in Mechanistic Interpretability of language models. For example, Elhage et al. (2021) takes the approach of analyzing graphs of underlying computation. Other approaches use synthetic datasets for benchmarking. Hyena Hierarchy Poli et al. (2023), for example, is evaluated using the following tasks on synthetic dataset: associative recall, majority, counting, ICL of functions, and arithmetic. LEGO Zhang et al. (2022) is a synthetic reasoning task that consists of a series of assignments and arithmetic operations involving variables. Nanda et al. (2023) investigate "grokking" through the task of modular addition. See also Pearce et al. (2023).

Our fundamental observation is that these tasks involve in-context learning. The number of tasks or operators involved (such as associative recall, majority, or arithmetic) are small. In contrast, real-world language understanding requires more operators, memorization of lot more facts, and reasoning over memorized facts. LoLaMeMe fills this gap. LoLaMeMe includes LOgical reasoning over MEmorized facts for MEchanistic interpretation. In addition, LoLaMeMe supports types, latent variables as well as noise which brings it closer to natural language.

### 2.2 Language Model Architectures

Transformers Vaswani et al. (2017) are a landmark contribution which made several language model architectures possible: encoder only Devlin et al. (2018), decoder only Radford et al. (2019), and encoder-decoder Lewis et al. (2020). To support longer context lengths, architectures such as Longformer Beltagy et al. (2020) have been proposed. A recent architecture for long-context lengths is Hyena Poli et al. (2023). Hyena is based on convolutions and does not use transformers and is attention-free. FNet hybrid model Lee-Thorp et al. (2022) introduced the idea that by only 2 layers of self-attention, the performance of non-attention model can be similar with the transformer model.

The role of transformers and convolutions in language understanding has to be explored. In this paper, we use LoLaMeMe framework and show architectures which interleave Transformer and Hyena blocks perform certain tasks well.

## 3 LoLaMeMe-Language Features

LoLaMeMe tries to mimic several aspects of natural language though it is couched in programming language syntax. Actually, there are few syntax variations and the variations are also mixed up to ensure such phenomenon in natural languages. We use programming language because it is easy to define the semantics of language constructs. We explore the following aspects of natural language through LoLaMeMe constructs.

### 3.1 Alphabet and Vocabulary

Natural languages have have a certain vocabulary size and words in the vocabulary have certain number of characters. We quantify these aspects through the number of characters of variable names and the number of variable names.

## 3.2 GRAMMAR

We use arithmetic expression like syntax for the "sentences" of the language. We use five operators: addition, subtraction, multiplication, division and modulo.

## 3.3 MEMORY

Memory is based on tokens not present in the prompt. In case of associative recall, the strings are part of training and the model has to retrieve the most appropriate result from *memory*. We use global variables, variable which have the same value in all expressions, as a measure of memory. We vary the number of global variables in the corpus.

## 3.4 LOGIC/REASONING

Logic or reasoning is based on a chain of reasoning such as the following:

$$a = 2; b = 1; c = (a/b) + X; e = c * Y; print(e);$$

The models are expected to interpret and execute the above sequence of expressions containing both local variables (in lower case, in this example) and global variables (in upper case, in this example). In other words, logical reasoning is performed on a sequence of arithmetic expressions. See Appendix B for examples of actual data.

## 4 TWO INSTANTIATIONS: LOLA AND MEME

We now describe two of LOLAMEME in programming language based syntax and semantics. The language supports variable names of specified sizes. The operators supported are addition, subtraction, multiplication, and modulo. The two languages, LoLa and MeMe, have differences in syntax as shown below.

| Operation | LoLa | MeMe |
|---|---|---|
| Addition | + | \| |
| Subtraction | − | ! |
| Multiplication | * | @ |
| Division | % | / |
| Grouping | () | {} |
| Case | Camel | Snake |

A module or paragraph consists of a sequence of roughly "STATEMENT COUNT" number of statements. The output of the module is the result of execution of the module. This is the semantics of the module.

Global Variables for Memory: Each corpus has 'GLOBAL VAR COUNT' number of global variables. The global variables are assigned the same value in the training corpus (at least "GLOBAL VARIABLES NUM APPEARANCE" times). These variables are used but not assigned in the test corpus with global variables.

Local Variables: These are used in module and it's values have scope limited to that module only.

Noise: In a module or paragraph, there can be statements which may not used to calculate the final result. These should be ignored.

Mixing: In the test corpus, we mix LoLa and MeMe constructs. We test which model generalizes better.

Variable Names: Variable names can be between "MIN VAR LENGTH" and "MAX VAR LENGTH" characters long. We generate local variable names and global variable names based on dataset configuration.

Latent Types: Variables have latent types which changes their value. Assume variables starting with `a-A` have `Type A` and those beginning with `A-B` have `Type B` and those beginning with `B-z` has

type C. Here `A`,`B` are a characters in `x-z` such that $a \leq A < B \leq z$. The probability of the type is $p$. The following is the rule used with latent types:

In the expression `var1 op var2`, if the type of `var1` is `Type A` and that of `var2` is `Type B`, then the effective value is `2*var1 op var2/2`.

Interpreter: In addition to data set creation, we need to create a simple interpreter which executes the code and outputs the result. We write the interpreter for only one language. We translate the other language as well as the mixed language to this language by string substitution. (It may be easier to convert snake case to camel case.)

All this is controlled through the dataset configuration show in in Appendix A.1. The dataset creation process is shown in Appendix D.1 and sample dataset is shown in Appendix B.

## 5 MODELS

Inspired by previous work Lee-Thorp et al. (2022), we propose the T HEX, which is the combination of transformer attention and Hyena Operator. More specifically, we replace certain layer of the Hyena (153M) model with the GPT-2 layer. Whlle other combinations are possible, we start with this simple hybrid model. Since the T HEX replaces one layer of the Hyena model to the GPT-2 layer, we use T HEX-$n$ to represent the T HEX which replaces the $n$-th layers counting from the input layer.

Considering that the dimensionality of the hidden states might influence the performance of the model, we set it to be 768 for GPT-2 layer and 864 for Hyena layer, and add linear layer for connecting the GPT-2 layer and Hyena Layer.

As baselines, we also test GPT-2 Radford et al. (2019)(124M) which is a decoder transformer model. Hyena (153M) Poli et al. (2023), which uses their proposed HyenaOperator which is a drop-in replacement for attention.

For all models, as we focus on testing its performance resulting from the architecture instead of the pre-training, all models are initialize randomly instead of using the pretraining weights.

## 6 EXPERIMENTS

We perform the following experiments.

**T HEX variations:** We vary the layer in the T HEX model which has the GPT-2 layer. Section 6.1

**Memorization:** We create test sets with and without global variables. The training set always has global variables. We vary the number of global variables in the training set from 100 to 1000 in steps of 100. See Section 6.2.

**Variable Length:** We vary the average number of characters in the variable name from 3 to 8. The actual size is $\pm 1$ or $\pm 2$ over the average. See Section 6.3.

**Long Input:** We increase the input size by increasing number of statements from 5 to 15. We experiment to see model's behaviour on longer more complex dataset. See Section 6.5.

**In-context learning:** We experiment to see if models can be taught to do in context learning. We do this by training the model on large dataset. See Section 6.4.

**Learn from Multiple languages:** We experiment to see if models learns aspects of multiple languages. We create a dataset with two languages and test the model in two different ways. See Section 6.6.

**Performance on public dataset:** We test models on Listops [Nangia & Bowman (2018)] dataset. See Section 6.7.

**Metrics:** In our experiments, we utilize exact match as the metric for evaluating models' performance.

## 6.1 T HEX VARIATIONS

In our experiments, the difficult of the tasks include two aspects: 1) the ability for logical reasoning; 2) the ability to remember the values of the variables used for calculating. Therefore, we create two test sets: 1) the global variables used for calculating is not provided which requires the model to remember the global variables during the training; 2) all the variables used for calculating is provided as the prompts, which focuses on the models' in-context learning ability.

In Table 1 , we show the performance of GPT-2, Hyena and T HEX. Since the T HEX replace one layer of the Hyena model to the GPT-2 layer, we use T HEX-n to represent the T HEX which replace the *n*th layers counting from the inputs. Our decision to choose n>=9 were based on early experiment described in Section D.

For the settings of including global variable shown in Table 1, both Hyena, and T HEX achieves higher performance than the GPT-2, which shows that Hyena Operator based model is better at synthetic reasoning task. Meanwhile, we observe that the most of the T HEX settings achieve better performance than the Hyena model (except for T HEX-16 and T HEX-17). This phenomenon shows that the introduction of the GPT-2 layer can increase the performance of Hyena Operator.

For the settings not including global variables, from Table 1, we observe that the performance of all models drop dramatically. This proves that compared with in-context learning, all these models are more likely to remember the parameters shown repeatedly in the training. Among all models, GPT-2 is the most stable one since its performance with the setting of including global variables is quite poor. When comparing Hyena and T HEX, we observe that all versions of T HEX achieve better performance than the Hyena, which shows that the introduction of GPT-2 layer can increase the in-context learning ability of the Hyena model.

| Model | With Global | Without Global |
|-------|-------------|----------------|
| Hyena | 0.1432 (0.0037) | 0.0033 (0.0007) |
| GPT-2 | 0.0066 (0.0010) | 0.0036 (0.0002) |
| T HEX-9 | 0.1663 (0.2247) | **0.0058 (0.0007)** |
| T HEX-10 | 0.2886 (0.0738) | 0.0046 (0.0014) |
| T HEX-11 | 0.2954 (0.0770) | 0.0045 (0.0015) |
| T HEX-12 | **0.3631 (0.0351)** | 0.0047 (0.0008) |
| T HEX-13 | 0.2932 (0.0785) | 0.0046 (0.0003) |
| T HEX-14 | 0.1652 (0.1475) | 0.0048 (0.0019) |
| T HEX-15 | 0.2163 (0.1826) | 0.0044 (0.0013) |
| T HEX-16 | 0.1372 (0.1164) | 0.0048 (0.0008) |
| T HEX-17 | 0.0298 (0.0214) | 0.0036 (0.0008) |

Table 1: The performance of GPT-2, Hyena and T HEX tested with LOLAMEME framework with the settings of including global variables and not including global variables. The results are reported with the LOLA datasets. In the experiments, we replace the 9-17 layers of Hyena. We report the mean and standard deviation (shown in the bracket) of exact match with various random seed.

With these two settings, we show that the combination of Hyena and GPT-2 can increase the in-context learning ability as well as keeping or even increasing models' ability for memorization. The results of MEME are shown in Appendix C.1.1 and the observations are similar to the results of LOLA reported here.

## 6.2 ARE MODELS GOOD AT MEMORIZATION?

The number of global variable is an important factor to influence the difficulty of LOLAMEME. Therefore, we conduct experiments with different number of global variables to analyze how global variable number influence the performance of the model. The dataset config for this experiment is shown in Appendix A.3. In addition to global variables, there are latent variables in the dataset which will test if the models are able to remember what the latent type meaning.

We show the performance of different models as a function of number of global variables in Figure 1 and Figure 2. It can be seen that Hyena model memorizes much better than GPT-2 model when the number of global variables are 100. When they are increased to 1000, the performance drops

and neither model memorizes anything. This could be due to the fact that the train dataset size is constant i.e. 100,000. This shows that hyena model learns much better than GPT-2 when dataset size is smaller. T HEX-13 in Figure 1 however performs better than rest on an average from 100-1000 global variables. This shows replacing hyena block with attention block in 13th layer has the highest impact.

In Figure 1 when number of global variables are 800 - 1000, attention in 15th layer becomes highest performing. In addition to this, when number of global variables is 700, there's a dip in scores for all models. We'd investigate this behaviour in future work.

These experiments are conducted for 3 times and we plot mean scores.

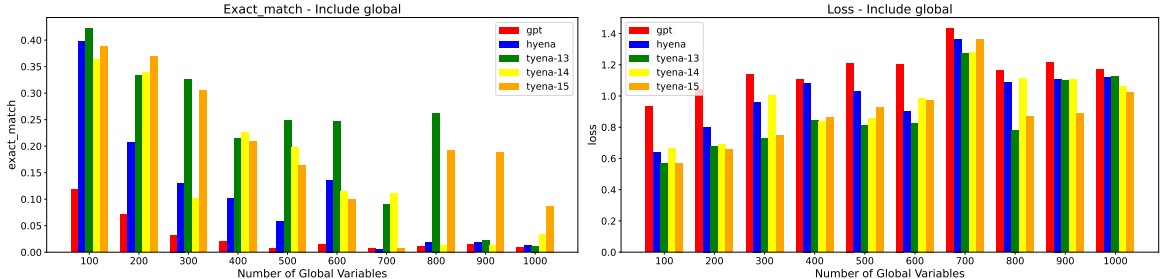

Figure 1: Memorization performance comparison between Hyena and GPT-2. This dataset includes global variables.

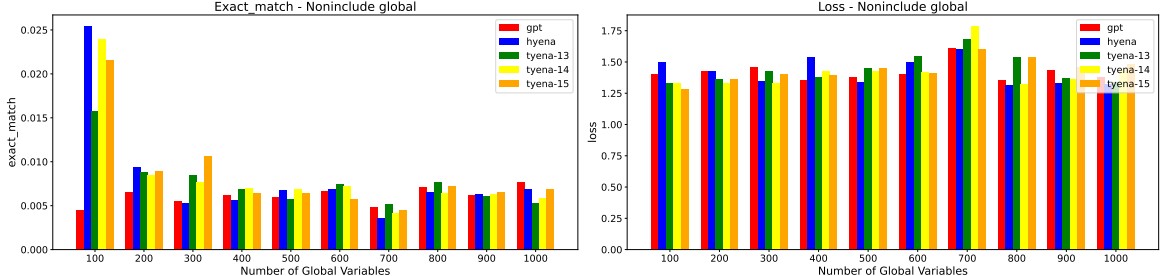

Figure 2: Memorization performance comparison between Hyena and GPT-2. This dataset does not includes global variables.

## 6.3 DOES LENGTH OF VARIABLES INFLUENCE MODEL PERFORMANCE?

In this experiment, we conduct experiments to show how the variable length influences the performance of different models. More specifically, we create multiple datasets with the mean length of variable as 3, 4, 5, 6, 7, 8, and 9. For each mean length, we set the interval of the variable length to be $[round(0.75 * M), round(1.25 * M)]$. We conduct experiments with and without global variables. The detailed settings are shown in Appendix A.2.

In Figure 3, we show the exact match for GPT-2, Hyena, and T HEX with the settings including global variables (left) and not including global variables (right) for the "LOLA". (The results of "MEME" are shown in Appendix C.1) Within the settings of including global variables, Hyena consistently outperforms GPT-2 across all mean variable lengths in terms of "exact match". *We observe that for most variable lengths, both the* T HEX-*13 and* T HEX-*15 achieve better performance than the Hyena model. This shows that the improvement of* T HEX *model for the settings with global variable is stable across all variable lengths.* We also notice that, with the increase of variable length, the performance of T HEX increases. This might because longer variable name are more likely to be split into multiple tokens which can help T HEX to remember the values of variables more correctly. Also we notice that T HEX-13 is a better choice than T HEX-15 which is in consist with the results reported before. These observations are also supported by the trends shown in the loss values in Figure 4.

For the settings of not including global variables shown in Figure 3 (right), Mirroring observations from settings with global variables, T HEX-13 achieves the best performance for most of the time. Being different from settings with global variable, Hyena is worse than the GPT-2 at most of the variable length. Given the minute values of the exact match, this anomaly may be attributed to random variations.

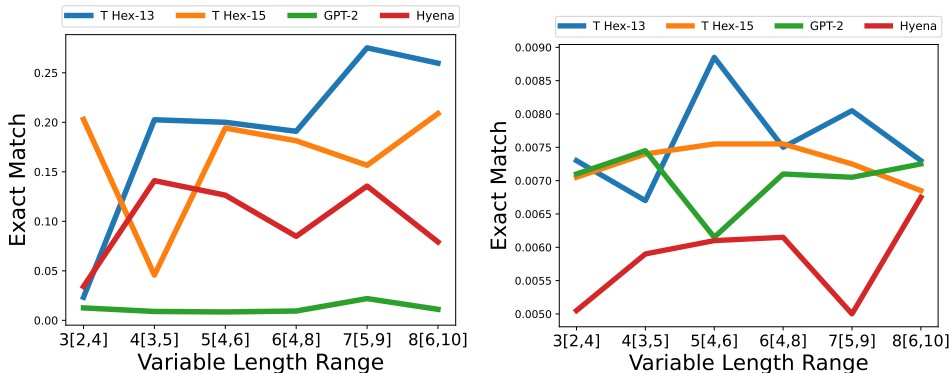

Figure 3: Exact match and loss for GPT-2 and Hyena with different variable Length. In figure, we provide the range of the variable length, and the mean length in the format of mean [min,max]. This is on dataset with (left) and without (right) global variables.

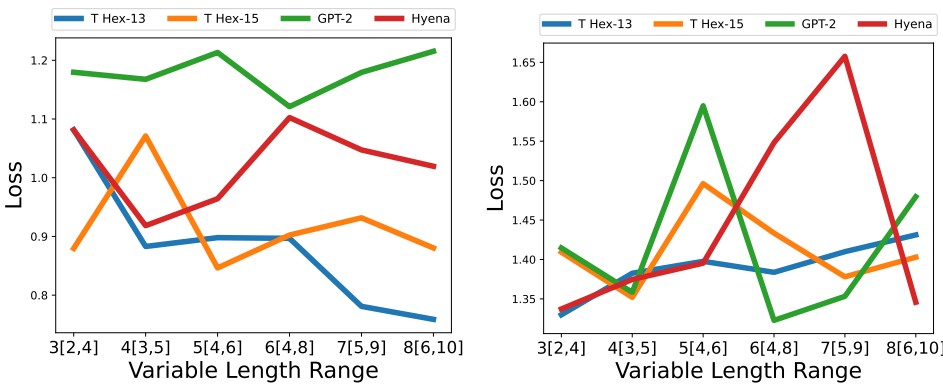

Figure 4: Loss for T HEX , GPT-2 and Hyena with different variable Length. In figure, we provide the range of the variable length, and the mean length in the format of mean [min,max]. This is on dataset with (left) and without (right) global variables.

### 6.4    DOES MODEL LEARN OPERATORS? (IN-CONTEXT LEARNING)

By comparing the performance shown in Table 1, we observe that all models (T HEX, Hyena and GPT-2) perform poorly on the test dataset without global variables. We assume this is because the models, especially T HEX, focus on remembering the global variables and ignore the learning of operators and local variables. Therefore, we create a dataset without global variables and use it to train all models and test the performance of non-global variables. In this setting, we train and evaluate model's ability for learning the operation and the in-context learning of local variables.

More specifically, we create a operator-pre-training dataset with 40,000,000 samples, and each sample includes 5 statements. The mean token number for each sample is about 100 tokens. For each model, we train 40,960,000 samples and the total training token number is 4 billion. We also create a evaluation and test sets in which there are 10000 samples for testing. In This experiment setting, for T HEX, we only test the T HEX-13 and T HEX-15 which achieves the best performance in previous experiments with and without global variables.

The result are show in Table 2. We observe that compared with the reported performance in Table 1 the performance of all models increase dramatically after operator-pre-training. This means that all models can learn operation and have the ability of in-context learning to remember the value of local variables. By comparing the performance of all models, we notice that the GPT-2 is significantly poorer than the other models. Meanwhile, the performance of T HEX-13 is the best among all models.

| Model | Without operator-pre-training | With operator-pre-training |
|---|---|---|
| Hyena | 0.0033 (0.0007) | 0.6726 (0.0239) |
| GPT-2 | 0.0036 (0.0002) | 0.1908 (0.0040) |
| T HEX-13 | 0.0046 (0.0003) | **0.6808 (0.0261)** |
| T HEX-15 | 0.0044 (0.0013) | 0.5785 (0.0029) |

Table 2: The exact match of models before and after operator-pre-training. The reported result is the mean of 3 rounds and the standard deviation is reported in the bracket.

## 6.5 MODEL BEHAVIOUR ON LONGER INPUT LENGTH

In this experiment, we increase the number of statement from 5 to 15 in basic config shown in Appendix A.1. The number of global variables is 1000 where the models performed worse in memorization experiment in Section 6.2. Therefore this dataset becomes even harder as there are 1000 global variables and the number of statements is increased by 10. With this increased complexity, we keep all the training setting same. The results are shown in Table 3.

Based on the results, we can see that all the models struggle to learn this long and more complex dataset. Hyena performs the best followed by T HEX-9. Evident from the pretraining in Section 6.4, more pre-training larger dataset would be required for better results. But based on current training, for longer input size, attention in layers $\leq 9$ shows better performance. T HEX-11 to T HEX-15 showed 0 loss after few epochs and showed 0 exact match. We would investigate this issue in our future work.

| Model | Include Global | Not Include Global |
|---|---|---|
| Hyena | 0.044 | 0.005 |
| GPT-2 | 0.005 | 0.003 |
| T HEX-1 | 0.010 | 0.002 |
| T HEX-3 | 0.017 | 0.004 |
| T HEX-5 | 0.012 | 0.003 |
| T HEX-7 | 0.008 | 0.002 |
| T HEX-9 | 0.027 | 0.004 |
| T HEX-11 | 0.000 | 0.00 |
| T HEX-13 | 0.000 | 0.00 |
| T HEX-14 | 0.000 | 0.00 |
| T HEX-15 | 0.000 | 0.00 |

Table 3: Model performance on longer input length.

## 6.6 DOES MODEL LEARN FROM MULTIPLE LANGUAGES?

For our framework, there are two languages: "LOLA" and "MEME". Although they have the same syntax, the symbol for the operators and the variable named rules are different. Therefore, one question is that if the model can only learn these two languages in different samples, what's the performance of it when combining these two languages in one sample. [2]

For training, we create a mixing dataset in in which half of the data is created based on "LOLA" and the other half is based on "MEME". For testing, we have two different settings: 1) sample-mixing: each sample is either "LOLA" or "MEME"; 2) module-mixing: the statements can be the combination of "LOLA" and "MEME" (e.g. zAb = (2@3}).

---

[2]The situation is to mimic the situation that when learning multiple languages, there is only one language in each sentence, but when using, multiple languages might show in one sentence.

| Model | sample-mixing | module-mixing |
|-------|---------------|---------------|
| GPT-2 | 0.249 | 0.246 |
| Hyena | 0.492 | 0.790 |
| T HEX-13 | **0.738** | **0.792** |
| T HEX-15 | 0.715 | 0.747 |

Table 4: Performance of all models with different language mixture settings. The reported results are the mean exact match of multiple rounds. The standard deviation is reported in the bracket.

In Table 4, we show the results of two mixture methods. As we can observe that GPT-2 still achieves the worst performance among all models similar to the single-language experiments. For the comparison between hyena and T HEX, T HEX-13 achieves the best performance. We can observe difference between Hyena and T HEX-13 is significantly bigger in sample-mixing but shrinks in module-mixing.

### 6.7 HOW IS PERFORMANCE ON EXISTING DATASETS?

**Listops**   We train all the models on Listops-1000 dataset for 1 epoch. The results are shown in Table 5. Based on the results, it can be seen that for longer input length as in Listops dataset, lower layers of attention in T HEX has better performance. And based on the results on LOLAMEME in Section 6.1, when the input length was comparatively smaller, attention in higher layers has more impact. This is in accordance with our observation in Section 6.5 in which for longer and more complex input, T HEXwith attention in lower layers showed better performance. This shows that LOLAMEME framework can effectively be used to mimic various aspects of language.

| | exact-match |
|--|-------------|
| GPT-2 | 0.361 |
| Hyena | 0.185 |
| **T HEX-4** | **0.362** |
| T HEX-1 | 0.360 |
| T HEX-13 | 0.321 |
| T HEX-15 | 0.276 |

Table 5: Performance on Listops-1000.

## 7   CONCLUSIONS

We have proposed a new framework, called LOLAMEME for mimicking different aspects of natural language. LOLAMEME can be used to test models on various aspects of language in a controlled fashion. We have instantiated LOLAMEME with two different manifestations, called LoLa and MeMe. We have shown how two different architectures, GPT-2 and Hyena, perform with respect to different aspects of language. We have also proposed a hybrid architecture, T HEX, that combines GPT-2 and Hyena blocks. T HEX outperforms GPT-2 and Hyena on most LOLAMEME tasks and on a related benchmark dataset. We believe ideas presented in this paper will inspire research on more systematic evaluations of large language models and on more effective model architectures.

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

# Appendix

## A   DATASETS AND THEIR CONFIGURATIONS

### A.1   BASECONFIG

```
baseconfig = {
    "LANGUAGE": "LOLA",
    "OPERATORS_LOLA": ['+', '-', '*', '%'], #operators used when LANGUAGE is LOLA
    "OPERATORS_MEME": ['|', '!','@','/'], #operators used when LANGUAGE is MEME
    "GROUPING_LOLA": ['(', ')'], #Grouping bracket style used when LANGUAGE is LOLA
    "GROUPING_MEME": ['{', '}'], #Grouping bracket style used when LANGUAGE is MEME
    "CASE_LOLA": 'camel',
    "CASE_MEME": 'snake',
    'VAR_NAME_UNDERSCORE_PROBABILITY':0.5, #Add underscore or capitalize a random char in VAR
            based on case
    'LATENT_CARACTER_ASCII_A':23, #0-23
    'LATENT_CARACTER_ASCII_B':-1, #23- [-1]
    'LATENT_CARACTER_ASCII_C':-1, #-1 . Here type B==type C
    'LATENT_VARIABLE_PROBABILITY':(0.3,0.4),#prob that LATENT_VARIABLE_PROBABILITY A is chosen
            is 0-0.3, type B is chosen is 0.3-(0.3+0.4) and type C is (0.3+0.4) - [-1]
            LATENT_VARIABLE_PROBABILITY times type B will be chosen
    'LATENT_TYPE_A_MODIFICATION':'*2',# if var stars with ascii b/w [0-23], var->var*2
    'LATENT_TYPE_B_MODIFICATION':'/2',# if var stars with any char in between ascii 23-26, var
            ->var/2
    'LATENT_TYPE_C_MODIFICATION':'/2',# if var stars with any char in between ascii 23-26, var
            ->var/2
    "GLOBAL_VAR_COUNT": 500, # number of unique global vars
    "LOCAL_VAR_COUNT": 10, # number of unique local vars
    "STATEMENT_COUNT": 5, # number of statements in one data item
    "EXPRESSION_MIN_DEPTH":1, #min recursions in a statement
    "EXPRESSION_MAX_DEPTH": 2, # max recursions in a statement
    "MIN_VAR_LENGTH": 3, #min string length of variables
    "MAX_VAR_LENGTH": 10, #max string length of variables
    "MIN_INT_VALUE": 1, #min values assigned of variables
    "MAX_INT_VALUE": 100, #max values assigned of variables
    "DATASET_SIZE":int(1e5), #train dataset size
    "TEST_DATASET_SIZE":int(1e4), #test dataset size
    "GLOBAL_VARIABLES_NUM_APPEARANCE":1000, #number of times each global var appears in
            dataset
    "USE_FAKER":False, #use faker to generate variables
}
}
```

### A.2   CONFIGFORVARIABLELENGTH

BaseConfig is used with following changes:

```
There are no latent variables used for this experiment.
'LATENT_TYPE_A_MODIFICATION':'',
'LATENT_TYPE_B_MODIFICATION':'',
'LATENT_TYPE_C_MODIFICATION':'',

The min and max variable lengths are controlled using these parameters for the experiment:
'MIN_VAR_LENGTH', 'MAX_VAR_LENGTH'
```

### A.3   CONFIGFORMEMORIZATION

The config for memorization is same as BaseConfig except for below parameter which is used to control number of global variables:

```
'GLOBAL_VAR_COUNT'
```

## B   EXAMPLE DATASET

The config used for this example dataset is shown in Appendix A.3.

## B.1 Training Dataset

A sample of Input from training dataset.

```
['tmrGk = 24;xnbolt = 77;xnbolt = (yI % aL);oZ = (dVfa - tmrGk);xnbolt =
    (84 % xvHMi);print(xnbolt)',
 'fIgLd = eTbZX;xnbolt = (76 * zjEy);oZ = (zG - 45);znmani = 8;fIgLd =
    yCHe;print(fIgLd)',
 'lsok = (dnmQTvq + wJyl);glkpJ = 34;xnbolt = ztjz;lsok = (zMig - mxMlpAi
    );xYBr = (pmxO + kgD);print(lsok)',
 'lsok = xqX;fIgLd = (yAW + pltjL);xAkt = (dJVgt + kNd);pBW = 56;fIgLd =
    mIW;print(fIgLd)',
 'yAW = 50;pcbZAu = (eTbZX + yMWB);lsok = kNd;bHJFL = vKa;bHJFL = (ybU %
    mxMlpAi);print(bHJFL)']
```

A sample of Output. In total there are 2506 unique output values in training set.

```
['84', '47', '-48', '86', '36']
```

A sample of Global Variables

```
['tmrGk = 24',
 'znmani = 8',
 'glkpJ = 34',
 'pBW = 56',
 'yAW = 50',
 'yumY = 92',
 'qqbsV = 33',
 'mH = 26',
 'vDU = 98',
 'grKwxk = 94',
 'yqfG = 20',
 'xiQu = 4',
 'ybU = 36',
 'xvsnsOp = 81',
```

## B.2 Test dataset without global variables

A sample of Input

```
['bHJFL = (98 + 26);xYBr = bHJFL;oZ = (xYBr + bHJFL);bHJFL = (67 * 96);
    print(xYBr)',
 'oZ = (3 - 61);xnbolt = (75 * 62);oZ = 10;baKfrFf = (oZ * 14);print(oZ)
    ',
 'xnbolt = 18;xYBr = (33 - xnbolt);lsok = xYBr;oZ = 41;print(lsok)',
 'xAkt = 18;xAkt = (60 + 36);baKfrFf = 83;oZ = 94;print(baKfrFf)',
 'baKfrFf = (60 - 18);lsok = baKfrFf;pcbZAu = (lsok % lsok);pcbZAu = 53;
    print(pcbZAu)']
```

A sample of Output. In total there are 1996 unique output values in test set.

```
['94', '13', '30615', '2', '2']
```

## B.3 Test dataset with global variables

A sample of Input

```
['pcbZAu = tBi;lsok = mZj;xYBr = (qhhFVb * vDU);baKfrFf = kDN;print(
    pcbZAu)',
 'fuB = (xaQeVwz % yMWB);pcbZAu = pmxO;fuB = (tBi % qklUazeme);xnbolt =
    znmani;print(fuB)',
 'xYBr = mxMlpAi;xYBr = mxMlpAi;oZ = (qhhFVb - yNM);bHJFL = tmrGk;print(
    xYBr)',
```

```
'xYBr = (qqbsV + vDU);xAkt = (wSTCa * ywP);xAkt = 76;pcbZAu = (ybU +
    rqnDmId);print(pcbZAu)',
'bHJFL = (pFzd * 91);fuB = (ehL - yNOkq);xAkt = (dCE % jvL);xnbolt =
    xaQeVwz;print(fuB)']
```

A sample of Output. In total there are 1047 unique output values in test set.

```
['18', '18', '90', '77', '-24']
```

## C  DOES LENGTH OF VARIABLES INFLUENCE MODEL PERFORMANCE? - APPENDIX

### C.1  RESULTS FOR MEME

#### C.1.1  T HEX VARIATIONS

In Table 6, we show the results of replacing different layer from the Hyena model with the MEME settings. Similar to the results we have for the LOLA shown in Section 6.1, Hyena and most of the T HEX(except for T HEX-13 and T HEX-17) achieve better performance than the GPT-2 on both settings of including and not including global variables. Meanwhile most of the T HEX achieves better performance than Hyena.

| Model | Include Global | Not Include Global |
|-------|----------------|--------------------|
| Hyena | 0.1243 (0.0240) | 0.0038 (0.0008) |
| GPT-2 | 0.0120 (0.0123) | 0.0035 (0.0008) |
| T HEX-9 | 0.1107 (0.1088) | 0.0044 (0.0007) |
| T HEX-10 | 0.1234 (0.1569) | 0.0050 (0.0010) |
| T HEX-11 | 0.0938 (0.1137) | 0.0040 (0.0008) |
| T HEX-12 | 0.1777 (0.1390) | 0.0038 (0.0001) |
| T HEX-13 | **0.2148 (0.0659)** | 0.0032 (0.0012) |
| T HEX-14 | 0.1511 (0.1295) | **0.0052 (0.0015)** |
| T HEX-15 | 0.1859 (0.1592) | 0.0046 (0.0012) |
| T HEX-16 | 0.0499 (0.0390) | 0.0037 (0.0010) |
| T HEX-17 | 0.0769 (0.0382) | 0.0030 (0.0006) |

Table 6: The performance of GPT-2, Hyena and T HEX tested with LOLAMEME framework with the settings of including global variables ("Include Global") and not including global variables ("Not include Globale"). The results are reported with the MEME datasets. In the experiments, we replace the 9-17 layers of Hyena. We report the mean and standard deviation (shown in the bracket) of exact match with various random seed.

## D  EARLY EXPERIMENT TO SELECT ATTENTION LAYERS

In the memorization experiment with 500 global variables, we replaced these hyena layers with attention to scan and then dive deep in that direction. The results are shown in Table 7. From this experiment, we found that higher layers showed high performance.

| Model | With Global | Without Global |
|-------|-------------|----------------|
| Hyena | 0.0092 | **0.0058** |
| GPT-2 | 0.0095 | 0.0056 |
| T HEX-0 | 0.0124 | 0.0045 |
| T HEX-1 | 0.2134 | 0.0052 |
| T HEX-9 | 0.2214 | 0.0041 |
| **T HEX-14** | **0.3369** | 0.0005 |

Table 7: Early Experiment to scan attention layers.

### D.1  Dataset Creation Process

The objective of our dataset creation process is to simulate a rich diversity of scenarios, encompassing different programming constructs, variable naming conventions, operators, and expression depths, all governed by a highly configurable system.

#### D.1.1  Configuration System

Our system is initiated with a configuration dictionary (config) (base config is shown in Appendix A.1), which contains various parameters to steer the data generation process.

#### D.1.2  Global Variable Generation

A specified number of global variables are created. Each variable name is generated based on the naming convention specified in the config. After ensuring uniqueness, each global variable is assigned a random integer value within a defined range.

#### D.1.3  Local Variable Generation

Depending on the configuration, a fixed number of local variables or a dynamic set can be generated. Similar to global variables, each local variable's name adheres to the naming convention and ensures no overlap with already generated names.

#### D.1.4  Statements and Depth

One of the core components of the dataset is the generation of programming constructs, or as we commonly refer to them, "statements". A statement in our dataset can range from simple assignments to more complex expressions involving multiple operators, variables, and nested subexpressions.

Depth is a crucial factor in determining the complexity of these statements. Depth refers to the number of nested sub-expressions within a single statement. For instance, a statement with a depth of 1 might look like a + b, while a depth of 2 could yield a + (b * c), and a depth of 3 could further nest as a + (b * (d - e)).

The config allows users to specify the minimum and maximum depths (EXPRESSION_MIN_DEPTH and EXPRESSION_MAX_DEPTH), ensuring a varied level of complexity across the dataset.

#### D.1.5  Input Item creation

One input item in dataset has "STATEMENT_COUNT" number of statements. Each statement is created one by one. The first variable is however is global variable or local variable using a value from constant or global variable or is function using global variables, constants and operators chosen randomly with some probability. Once the first variable for statement is created, rest of the statement is made by using this i.e. depth is recursively added. All the variables used in statements are tracked to use in future statements. It is made sure that no unassigned variable is used to assign to a variable. Only variables which are assigned a value are allowed to be used for assignment or in a function.

Future statements are created using variables randomly chosen between tracked local variables, global variables, constants or new local variables are created.

Total of "STATEMENT_COUNT" number of statements are created like this and at the end one of the randomly chosen tracked variables is printed. We convert the input item to python language and use python interpreter to generate output and that becomes the label. This makes input and label pair. "DATASET SIZE" number of unique input output pairs are then created to make a dataset.

