# OpenReview forum: "LOLAMEME: LOGIC, LANGUAGE, MEMORY, MECHANISTIC FRAMEWORK"
_ICLR.cc/2024/Conference — Submitted to ICLR 2024_

### Official Review · Reviewer_JYqN · 2023-11-01

**Soundness:** 2 fair
**Presentation:** 1 poor
**Contribution:** 2 fair
**Rating:** 3
**Confidence:** 3

**Summary:**

This paper proposes a novel framework called LOLAMEME including two instantiations, the LoLa language, and MeMe languages. Then it introduces a hybrid architecture T HEX and compares it with transformer-based GPT-2 and convolution-based Hyena based on the LOLAMEME framework. Furthermore, this work conducts comprehensive experiments to demonstrate the effectiveness of T HEX in different tasks.

**Strengths:**

1. The new framework LOLAMEME similar to natural language is impressive and interesting.
2. This work builds multiple datasets with several billion tokens based on the LOLAMEME framework, which would contribute to future research.
3. This work performs comprehensive experiments over these datasets and a related benchmark dataset to show the effectiveness of the new framework.

**Weaknesses:**

1. The motivation for the model design is not clearly discussed in this work. I am confused about the differences among T HEX, GPT-2, and Hyena.
2. The structure of this paper is not clear enough, which is very hard to follow.
3. I would suggest that an illustration figure be provided to clearly show the main idea of the LOLAMEME framework, which will make this work easier to understand.
4. Some sentences should be revised and the format should be unified. For instance, in the Abstract, "We extend current mechanistic schemes to incorporate Logic, memory, and nuances of Language such as latent structure", I am curious why the first letter of Logic and Language in this sentence are capitalized.

**Questions:**

1. What are the differences among T HEX, GPT-2, and Hyena?
2. What are the advantages of T HEX?
3. Why not provide an illustration figure to show the model framework?

---

> ### Author Response · Authors · 2023-11-23
>
> Thank you for your thorough review and valuable feedback on our paper. We appreciate the opportunity to address your concerns and clarify some aspects of our work.
>
> 1. What are the differences among T HEX, GPT-2, and Hyena?
>     1. In the paper, we use GPT-2 model which uses attention operator, Hyena model which uses hyena operator and T-Hex which is similar to hyena model in which 1 layer of hyena operator is replaced with attention operator.
> 2. What are the advantages of T HEX?
>     1. Using LOLAMEME framework, we tested the GPT-2 and hyena model architectures. We identified various areas where GPT-2 using attention operator works better and Hyena using hyena operator works better. Guided by the experiment results using LOLAMEME framework, we show that attention and hyena have complementary strengths using T-HEX. Using attention operator together with hyena operator, boosts the model architecture’s performance on various aspects such as memorization, in-context learning, language mixing, longer input length and more listed in the paper.
>     2. We show T-Hex performs better than Hyena and GPT-2 on various aspects of the language and we also show results on existing dataset “Listops”. Using LOLAMEME framework, better architectural design changes can be developed and tested on various aspects of the language.
> 3. Why not provide an illustration figure to show the model framework?
>     1. Thanks for the feedback, we will definitely add model architecture as well as LOLAMEME framework illustration in the next revision. This is the anonymous link to a draft of LOLAMEME framework illustration:  https://ibb.co/qWCPRnW . We will add a revised version to the paper.

---

### Official Review · Reviewer_5Z2e · 2023-11-02

**Soundness:** 1 poor
**Presentation:** 1 poor
**Contribution:** 1 poor
**Rating:** 3
**Confidence:** 3

**Summary:**

This paper talks a lot about mechanistic interpretability and evaluation of models with different architectures on synthetic benchmarks to generate more understanding of how they work. I fail to understand completely what insight is gained here.

**Strengths:**

.

**Weaknesses:**

The only changes done to the transformer architecture is to replace a single layer by a layer from the hyena model. The variations include only replacing a different layer of the transformer with the same hyena layer. Lots of experiments are done to compare the performance of variants and measure the impact on the quality under different input lengths, on some synthetic datasets, etc. But I don't see any insight that could be won from these experiments.

**Questions:**

None.

---

> ### Author Response · Authors · 2023-11-23
>
> Thank you for your thorough review and valuable feedback on our paper. We appreciate the opportunity to address your concerns and clarify some aspects of our work.
>
>
> 1. Natural languages has several aspects to it and it has become extremely difficult to test a model architecture on one or a set of aspects of the natural language. LOLAMEME framework is built to evaluate model architectures on various aspects of language such as grammer, vocabulary, logic/reasoning, memorization, multiple languages and more listed in the paper.
> 2. Using LOLAMEME framework, we tested the GPT-2 and hyena model architectures. We identified various areas where GPT-2 using attention operator works better and Hyena using hyena operator works better. Guided by the experiment results using LOLAMEME framework, we show that attention and hyena have complementary strengths using T-HEX. Using attention operator together with hyena operator, boosts the model architecture’s performance on various aspects such as memorization, in-context learning, language mixing, longer input length and more listed in the paper.
> 3. We show T-Hex performs better than Hyena and GPT-2 on various aspects of the language and we also show results on existing dataset “Listops”. Using LOLAMEME framework, better architectural design changes can be developed and tested on various aspects of the language.

---

### Official Review · Reviewer_iHzK · 2023-11-07

**Soundness:** 2 fair
**Presentation:** 2 fair
**Contribution:** 2 fair
**Rating:** 3
**Confidence:** 3

**Summary:**

This article introduces a novel framework called LOLAMEME that expands current mechanistic schemes to incorporate Logic, memory, and nuanced aspects of Language, such as latent structure. By using this framework, the authors compare three generative language model architectures: GPT-2 (transformer-based), Hyena (convolution-based), and proposed hybrid architecture T HEX which are constructed by replacing certain layer of the Hyena model with the GPT-2 layer. To instantiate LOLAMEME, the authors introduce two different manifestations, LoLa and MeMe, and evaluate the performance of the architectures across various aspects of language. The findings demonstrate that T HEX surpasses GPT-2 and Hyena on select tasks as well as a related benchmark dataset.

**Strengths:**

This work proposes a new hybrid architecture based on transformer-based GPT-2 and convolution-based Hyena. Experiments demonstrate the superiority of this architecture.

**Weaknesses:**

•	The motivation and problem formulation of this work is unclear. And the novelty and contribution of this paper are somewhat limited. The proposed new architectures are simply constructed by replacing certain layers of the Hyena model with the GPT-2 layer. Although some experiments demonstrate better performance on the proposed two test datasets, there may be a lack of validation experiments on other existing datasets. Additionally, providing some interesting findings or interpretations about the experiments through a deeper analysis of the proposed architectures should be better.
•	The construction procedure of the two datasets should be explained more clearly, and deeper consideration should be given to whether the experimental settings can reliably reflect the behavior of the related models, such as memorization and in-context learning.
•	There are quite a few typo errors, including grammar and table issues, in this paper. For example, in the abstract, it states "We propose the hybrid architecture T HEX and use LOLAMEME framework is used to compare three architectures." There are also typo errors in tables 3, 4, and 5. The grammar, figures, and tables in this paper may require some polishing.
•	The related work on mechanistic interpretability is not comprehensive. In fact, there is a considerable amount of work such as [1], [2], [3] attempting to interpret and understand the mechanisms of LLMs.
•	In section 6.5, it is unclear why TH EX-11 to T HEX-15 showed a loss of 0 after a few epochs but showed an exact match of 0. Further clarification or explanation is needed for this inconsistency.
[1] A Mathematical Framework for Transformer Circuits
[2] Interpretability in the Wild: a Circuit for Indirect Object Identification in GPT-2 small
[3] Investigating Gender Bias in Language Models Using Causal Mediation Analysis

**Questions:**

Please see the weaknesses.

---

> ### Author Response · Authors · 2023-11-23
>
> Thank you for your thorough review and valuable feedback on our paper. We appreciate the opportunity to address your concerns and clarify some aspects of our work.
>
> Addressing Strengths:
> We thank the reviewer for recognizing the potential of one of our contributions new hybrid architecture. Although , our contribution also include framework called LOLAMEME which is a mechanistic framework closer to natural language. LOLAMEME framework is used to explore the capabilities of different language model architectures, through which we show various findings of GPT-2, Hyena and show that in T-HEX architecture both attention and hyena operator are complementary to each other and show greater performance on various aspects of language.
>
> Addressing Weaknesses:
>
>
> 1. The motivation and problem formulation of this work is unclear. Natural languages has several aspects to it and it has become extremely difficult to test a model architecture on one or a set of aspects of the natural language. LOLAMEME framework is built to evaluate model architectures on various aspects of language such as grammer, vocabulary, logic/reasoning, memorization, multiple languages and more listed in the paper.
> 2. The proposed new architectures are simply constructed by replacing certain layers of the Hyena model with the GPT-2 layer. Although some experiments demonstrate better performance on the proposed two test datasets, there may be a lack of validation experiments on other existing datasets. Additionally, providing some interesting findings or interpretations about the experiments through a deeper analysis of the proposed architectures should be better.
>     1. Using LOLAMEME framework, we tested the GPT-2 and hyena model architectures. We identified various areas where GPT-2 using attention operator works better and Hyena using hyena operator works better. Guided by the experiment results using LOLAMEME framework, we show that attention and hyena have complementary strengths using T-HEX. Using attention operator together with hyena operator, boosts the model architecture’s performance on various aspects such as memorization, in-context learning, language mixing, longer input length and more listed in the paper.
>     2. We show T-Hex performs better than Hyena and GPT-2 on various aspects of the language and we also show results on existing dataset “Listops”. Using LOLAMEME framework, better architectural design changes can be developed and tested on various aspects of the language.
> 3. The construction procedure of the two datasets should be explained more clearly, and deeper consideration should be given to whether the experimental settings can reliably reflect the behavior of the related models, such as memorization and in-context learning. Thank you for your feedback, we will explain the dataset more clearly in the next revision.
> 4.  There are quite a few typo errors, including grammar and table issues, in this paper. For example, in the abstract, it states "We propose the hybrid architecture T HEX and use LOLAMEME framework is used to compare three architectures." There are also typo errors in tables 3, 4, and 5. The grammar, figures, and tables in this paper may require some polishing. Thank you for your feedback, we will address this in the next revision.
> 5. The related work on mechanistic interpretability is not comprehensive. In fact, there is a considerable amount of work such as [1], [2], [3] attempting to interpret and understand the mechanisms of LLMs.  We realize the importance of a comprehensive review of related work. Our paper will be updated to include a more exhaustive list of research in the domain of LLMs interpretability, referencing works such as [1], [2], [3] cited by the reviewer
> 6. In section 6.5, it is unclear why TH EX-11 to T HEX-15 showed a loss of 0 after a few epochs but showed an exact match of 0. Further clarification or explanation is needed for this inconsistency. One intuition for this is that, attention in higher layers requires more training data than in lower layers for more complex dataset. In section 6.4 we observed pre-training on larger dataset boosted the performance. In future work, we will address this with more research.

---

### Meta-Review · Area_Chair_qrzm · 2023-12-12

**Metareview:**

This paper presents a novel framework called LOLAMEME, which includes two instantiations: the LoLa language and MeMe languages. This framework expands current mechanistic schemes to incorporate logic, memory, and nuanced aspects of language. The paper introduces a hybrid architecture called T HEX, constructed by replacing certain layers of the Hyena model with GPT-2 layers. Comprehensive experiments demonstrate the effectiveness of T HEX in various tasks. However, the motivation for the model design is not clearly presented in the paper. Additionally, several reviewers have pointed out that the presentation of the paper needs significant improvement.

**Justification For Why Not Higher Score:**

The motivation for the model design is not clearly presented in the paper. Additionally, several reviewers have pointed out that the presentation of the paper needs significant improvement.

**Justification For Why Not Lower Score:**

N/A

---

### Decision · Program_Chairs · 2024-01-16

Reject